# Developing the Ascorbic Acid Test: A Candidate Standard Tool for Characterizing the Intrinsic Reactivity of Metallic Iron for Water Remediation

Xuesong Cui [1], Minhui Xiao [1,2], Ran Tao [1], Rui Hu [2], Hans Ruppert [3], Willis Gwenzi [4,5] and Chicgoua Noubactep [1,6,7,8,*]

1 Angewandte Geologie, Universität Göttingen, Goldschmidtstraße 3, D-37077 Göttingen, Germany; xiaominhui@hhu.edu.cn (M.X.)
2 School of Earth Science and Engineering, Hohai University, Fo Cheng Xi Road 8, Nanjing 211100, China
3 Department of Sedimentology & Environmental Geology, University of Göttingen, Goldschmidtstraße 3, D-37077 Göttingen, Germany
4 Grassland Science and Renewable Plant Resources, Faculty of Organic Agricultural Science, University of Kassel, Steinstrasse 19, D-37213 Witzenhausen, Germany
5 Leibniz Institute for Agricultural Engineering and Bioeconomy (ATB), Max-Eyth-Allee 100, D-14469 Potsdam, Germany
6 Department of Water and Environmental Science and Engineering, Nelson Mandela African Institution of Science and Technology, Arusha P.O. Box 447, Tanzania
7 Faculty of Science and Technology, Campus of Banekane, Université des Montagnes, Bangangté P.O. Box 208, Cameroon
8 Centre for Modern Indian Studies (CeMIS), Universität Göttingen, Waldweg 26, D-37073 Göttingen, Germany
* Correspondence: cnoubac@gwdg.de

**Abstract:** Granular metallic iron ($gFe^0$) materials have been widely used for eliminating a wide range of pollutants from aqueous solutions over the past three decades. However, the intrinsic reactivity of $gFe^0$ is rarely evaluated and existing methods for such evaluations have not been standardized. The aim of the present study was to develop a simple spectrophotometric method to characterize the intrinsic reactivity of $gFe^0$ based on the extent of iron dissolution in an ascorbic acid (AA—0.002 M or 2 mM) solution. A modification of the ethylenediaminetetraacetic acid method (EDTA method) is suggested for this purpose. Being an excellent chelating agent for $Fe^{II}$ and a reducing agent for $Fe^{III}$, AA sustains the oxidative dissolution of $Fe^0$ and the reductive dissolution of $Fe^{III}$ oxides from $gFe^0$ specimens. In other words, $Fe^0$ dissolution to $Fe^{II}$ ions is promoted while the further oxidation to $Fe^{III}$ ions is blocked. Thus, unlike the EDTA method that promotes $Fe^0$ oxidation to $Fe^{III}$ ions, the AA method promotes only the formation of $Fe^{II}$ species, despite the presence of dissolved $O_2$. The AA test is more accurate than the EDTA test and is considerably less expensive. Eight selected $gFe^0$ specimens (ZVI1 through ZVI8) with established diversity in intrinsic reactivity were tested in parallel batch experiments (for 6 days) and three of these specimens (ZVI1, ZVI3, ZVI5) were further tested for iron leaching in column experiments (for 150 days). Results confirmed the better suitability (e.g., accuracy in assessing $Fe^0$ dissolution) of the AA test relative to the EDTA test as a powerful screening tool to select materials for various field applications. Thus, the AA test should be routinely used to characterize and rationalize the selection of $gFe^0$ in individual studies.

**Keywords:** groundwater remediation; intrinsic reactivity; iron dissolution; permeable reactive barrier; zero-valent iron

## 1. Introduction

Metallic iron ($Fe^0$) is an effective reactive medium for the remediation of aqueous systems (e.g., groundwater, rainwater, stormwater, wastewater) polluted with numerous species, including chlorinated solvents, trace metals, nutrients and pathogens [1–10]. The standard redox potential of the $Fe^{II}/Fe^0$ electrode reaction ($E^0 = -0.44$ V; Equation (1))

makes $Fe^0$ theoretically an effective reducing agent for many reducible contaminants with $E^0 > -0.44$ V [11]. The introduction of $Fe^0$ as an efficient reactive material for subsurface permeable reactive barriers (PRBs) was based on this premise [11–14]. However, it was established long before the advent of $Fe^0$ PRBs that under environmental conditions, only protons ($H^+$ from $H_2O$ dissociation) can oxidize $Fe^0$ by an electrochemical mechanism (Equation (1)) [15,16].

$$Fe^0 + 2H^+ \Rightarrow Fe^{2+} + H_2 \qquad (1)$$

Recent overview articles on "$Fe^0$ for groundwater remediation" point out that further development of this promising technology is impaired by controversies on the operating mode of $Fe^0$ [9,17–19]. In fact, the view that $Fe^0$ is a generator of contaminant scavengers (e.g., solid iron corrosion products—FeCPs), and secondary reducing agents (e.g., $Fe^{II}$, $H_2$, $Fe_3O_4$), as summarized in Hu et al. [17], and supported by the seminal work of Whitney [15] has been reported to be "isolated misconceptions" [9,18].

Regarding granular $Fe^0$ (micro-scale) as an environmental reducing agent has created a circular reasoning that dragged the research community into an unprecedented confusion [9,17,18,20]. In particular, to enhance the efficiency of $Fe^0$ to remove selected contaminants from the polluted waters, bimetallic materials (e.g., $Fe^0/Cd^0$, $Fe^0/Ni^0$, $Fe^0/Pd^0$, $Fe^0/Pt^0$) and nano-scale materials (nano-$Fe^0$ and nano-bimetallics) have been developed [4,10,21,22]. However, the discussion of the reactivity is still based on the value of $E^0 = -0.44$ V for the electrode reaction $Fe^{II}/Fe^0$ [18,21,22]. One problem has been that the terms "efficiency" and "reactivity" have been often randomly interchanged [18,23–25]. In essence, the reactivity of any $Fe^0$ sample is fixed by the value $E^0 = -0.44$ V, while its efficiency is the "expression" or the "manifestation" of this reactivity ($Fe^0$ type or $Fe^0$ quality) as influenced by operating conditions (e.g., $Fe^0$ dosage and grain size, co-solutes, contaminant concentration, pH value, temperature). Clearly, under appropriate experimental conditions, a double amount ($2 m_0$) of a given $Fe^0$ material (fixed reactivity) is theoretically expected to remove more contaminant from an aqueous phase than a single amount ($m_0$). In other words, in theory, the efficiency of a $Fe^0$ material for water treatment depends (also) on the used dosage. However, in reality, doubling of iron dosage does not double the amount of contaminant removed. Thus, there is no linear relationship between the iron dosage in the system and the amount adsorbed and co-precipitated contaminants.

Enhanced remediation efficiency by varying the $Fe^0$ dosage is justified by variability of the surface area available for $Fe^0$ dissolution which is thought to be coupled to contaminant reductive transformations [11,21,22,26,27]. The higher efficiency of nano-$Fe^0$ relative to granular $Fe^0$ is based roughly on the same principle i.e., larger available surface area, despite difference in intrinsic reactivity [22,28–30]. Two questions arise: (i) Why do $Fe^0$ materials of similar sizes (e.g., nano-$Fe^0$) react differently if their reactivity is controlled by the same $E^0$ value? (ii) How can a contaminant ($E^0 > -0.44$ V) be reduced to a certain extent by a given $Fe^0$ sample and not at all by another $Fe^0$ material under the same operating conditions? It seems obvious that these two questions have not yet received the attention they deserve. Certainly, the thermodynamics (relative $E^0$ values) do not have the ultimate control, but the kinetics of iron dissolution is a key factor [31,32]. The fundamentals of electrochemical reactions teach that $Fe^0$ (uncorroded metal) is only one of the four components necessary for electron transfer. The other three components are: (i) an anode where $Fe^0$ is dissolved, (i) a cathode where released electrons are exchanged with a relevant species, and (iii) an electrolyte that transports $Fe^{2+}$ ions away from the anode. Uncorroded $Fe^0$ ensures the transfer of electrons from the anode to the cathode. There are two reasons why electrons cannot be exchanged between $Fe^0$ and dissolved contaminants: (i) $Fe^0$ is permanently shielded by a non-conductive oxide scale, and (ii) the oxide scale acts as a diffusion barrier for contaminants and a conduction barrier for electrons [17]. The present work focuses on $Fe^0$ as a stand-alone parameter for the efficiency of $Fe^0/H_2O$ systems.

The characterization of the actual contribution of $Fe^0$ to the process of contaminant removal using $Fe^0/H_2O$ systems is complicated by the complex interactions between relevant contaminants, operating conditions (e.g., initial concentration, stirring intensity), water

chemistry (e.g., availability of co-solutes, contamination level, pH value), and transport phenomena (e.g., advection, diffusion) in the bulk solution and in the vicinity of $Fe^0$ [21,33–35]. Despite the wide application of $Fe^0$ (e.g., bimetals, $Fe^0$, nano-$Fe^0$) for the treatment of a variety of pollutants, the evaluation of $Fe^0$ reactivity has not been standardized [21,34–40]. Of the few characterization tools presented for the $Fe^0$ intrinsic reactivity, only two are truly contaminant independent as they quantify the kinetics of $Fe^0$ dissolution (Equation (1)) by monitoring either dissolved Fe [34,41,42] or generated $H_2$ [38,43]. The most affordable method presented to date is the o-phenanthroline (Phen) method which uses only a spectrophotometer for $Fe^{II}$ determination and relies on $Fe^0$ dissolution in a dilute solution of o-phenanthroline (2 mM or 0.002 M) [42]. This simple colorimetric assay is certainly versatile for universal application, even in poorly-equipped laboratories lacking advanced analytical instrumentation. However, the toxicity of o-phenanthroline [43,44] and its higher price (229.00 €/100 g, www.sigmaaldrich.com—accessed on 28 March 2023) compared to ascorbic acid (vitamin C—8.00 €/kg, www.amazon.de—accessed on 28 March 2023) have motivated the development of the ascorbic acid test as a low-cost and environmentally friendly alternative.

Ascorbic acid (AA) is a well-known chemical reducing agent [44–49] that has also been used in environmental remediation [50–55]. AA has also been applied as an alternative to conventional reducing reagents (e.g., hydroxylamine, hydroquinone) in the standard method of converting iron(III) to iron(II) prior to forming a red-orange complex with o-phenanthroline in the spectrophotometric determination of iron [44,46]. $Fe^{II}$-AA complexes are extremely stable at pH values 3.0 to 8.0 [56,57]. In the present study, AA was chosen as a complexing agent to stabilize $Fe^{2+}$ from Equation (1) and sustain $Fe^0$ oxidative dissolution. The AA test requires one chemical reagent less than the EDTA method. AA is also nontoxic to researchers, environmentally friendly, and less expensive than EDTA or Phen [53,58].

This study presents a simple colorimetric assay to characterize the intrinsic reactivity of $gFe^0$, using ascorbic acid (AA) to complex $Fe^{2+}$ from $Fe^0$ oxidative dissolution (Equation (1)). Eight different $gFe^0$ samples were tested for Fe dissolution in a 2 mM AA solution to demonstrate the applicability of the method to assess their intrinsic reactivity. In addition, experiments with 2 mM EDTA and 2 mM Phen solutions were performed for comparison.

## 2. Materials and Methods

### 2.1. Solutions

Working solutions were prepared from L-hexuronic acid ascorbic acid (AA) (Wasser Hygiene Chemie GmbH, Hilgertshausen, Germany), disodium salt of ethylenediaminetetraacetic (EDTA) (AppliChem GmbH, Darmstat, Germany), and monohydrated 1,10-Phenanthroline (Phen) (Acros Organics, Geel, Belgium). An iron standard solution (1000 mg $L^{-1}$) (Fisher Scientific UK Limited, Loughborough, UK) was used to calibrate the used UV/VIS spectrophotometer. The Phen solution (0.2 M) and a buffer solution formed by mixing AA and sodium ascorbate (FeelWell GmbH, Gnarrenburg, Germany) were used to determine the aqueous iron concentration. All used chemicals were of analytical grade.

The pH values of the working solutions (2 mM each) were: ascorbic acid 3.1; EDTA 4.7; and Phen 8.3. Tap water from the city of Göttingen (Germany) was used to prepare the solution. Its pH value was 7.8.

### 2.2. Iron Materials

A total of 8 $Fe^0$ materials were selected and used in this study. The selection was based on their differential reactivity as determined in previous work [42,43]. The used $Fe^0$ materials had different geometrical shapes and size. Three of these were commercially available $Fe^0$ materials for groundwater remediation referred to as: (i) ZVI1 is material from iPutec GmbH Rheinfelden, Germany: (ii) ZVI2 and ZVI3 is directly reduced sponge iron material (DRI) from ISPAT GmbH, Hamburg, Germany; (iii) ZVI6: Connelly Iron from Connelly-GPM Inc., Chicago, IL, USA. ZVI4 and ZVI5 is scrap iron from the metal recycling

company Metallaufbereitung Zwickau/Germany. ZVI5 is a mixture of mild steels from various sources; ZVI4 is a mixture of cast irons. ZVI7 and ZVI8 are spherical iron samples from the Chinese company Tongda Alloy Material Factory.

Table 1 summarizes the main characteristics of the 8 $Fe^0$ samples along with their iron content as specified by the supplier. ZVI1, ZVI3 and ZVI5 were tested in column leaching experiments.

**Table 1.** Code and main characteristics of tested $Fe^0$ materials according to the supplier. n.s. = not specified; granular = mechanically broken pieces; sponge = particles with pitted surfaces; scrap = waste generated in any form: spherical = standard sphere with a smooth surface.

| Code | Shape | Size | Color | Specific Surface Area | Fe | Supplier |
|------|-------|------|-------|----------------------|-----|----------|
|      |       | (mm) |       | $(m^2/g)$            | (%) |          |
| ZVI1 | granular  | 0.05–5.00 | black | n.s.      | n.s.  | iPutec GmbH |
| ZVI2 | sponge    | 0.68–1.00 | black | n.s.      | 90.0  | ISPAT GmbH |
| ZVI3 | sponge    | 1.00–2.00 | black | n.s.      | 90.0  | ISPAT GmbH |
| ZVI4 | scrap     | 0.05–5.00 | black | n.s.      | n.s.  | Metallaufbereitung Zwickau |
| ZVI5 | scrap     | 0.05–2.00 | black | n.s.      | n.s.  | Metallaufbereitung Zwickau |
| ZVI6 | granulate | 0.05–10.0 | black | n.s.      | n.s.  | Connelly |
| ZVI7 | spherical | 0.05–1.00 | grey  | 0.74–1.26 | 99.99 | Tongda Alloy Material Factory |
| ZVI8 | spherical | 2.00      | grey  | 0.39      | 99.99 | Tongda Alloy Material Factory |

### 2.3. Experimental Methods

#### 2.3.1. Batch Experiments

Iron dissolution experiments were performed using 0.10 g of the ZVI1 sample in 50 mL of the three complexing agent: AA, EDTA, and Phen (each 2 mM) for up to 144 h (6 days). Further experiments using AA were performed for the 8 tested $Fe^0$ samples. All experiments were performed with the conventional quiescent (non-agitated, non-stirred) experimental protocol described in detail in earlier papers [41]. The experimental vessels were protected from direct sunlight and atmospheric dust. Each experiment was performed in triplicate and the average results are presented.

#### 2.3.2. Column Experiments

1.0 g of each $Fe^0$ material (ZVI1, ZVI3, and ZVI5) was placed in a chromatographic column with sand in the lower third and the 2 mM AA solution in the upper two thirds (Figure 1). $Fe^0$ was leached daily for five consecutive days (Monday through Friday) every week with about 180 mL of the AA solution (pH = 3.1) at temperature of 25 ± 4 °C. At each leaching event, the exact volume of the leachate was monitored and its iron concentration determined. The experiment was terminated when the leaching rate of the reactive $Fe^0$ material reached 40%.

### 2.4. Analytical Method

Analysis for iron was performed by using the phenanthroline method described in detail in earlier papers [42]. Although Fe(AA) exists already in the Fe(II) form, reduction by AA addition was performed just to follow the analytical protocol which includes also the calibrating solutions. Iron concentrations were determined using a Varian Cary 50 Scan UV-VIS Spectrophotometer (Cary instruments, LabMakelaar Benelux B.V., Zevenhuizen, The Netherlands) at a wavelength of 510.0 nm using a 1.0 cm glass cuvette. The instrument has been calibrated for iron concentrations $\leq 10$ mg $L^{-1}$. The pH values were measured with combined glass electrodes (WTW Co., Weinheim, Germany).

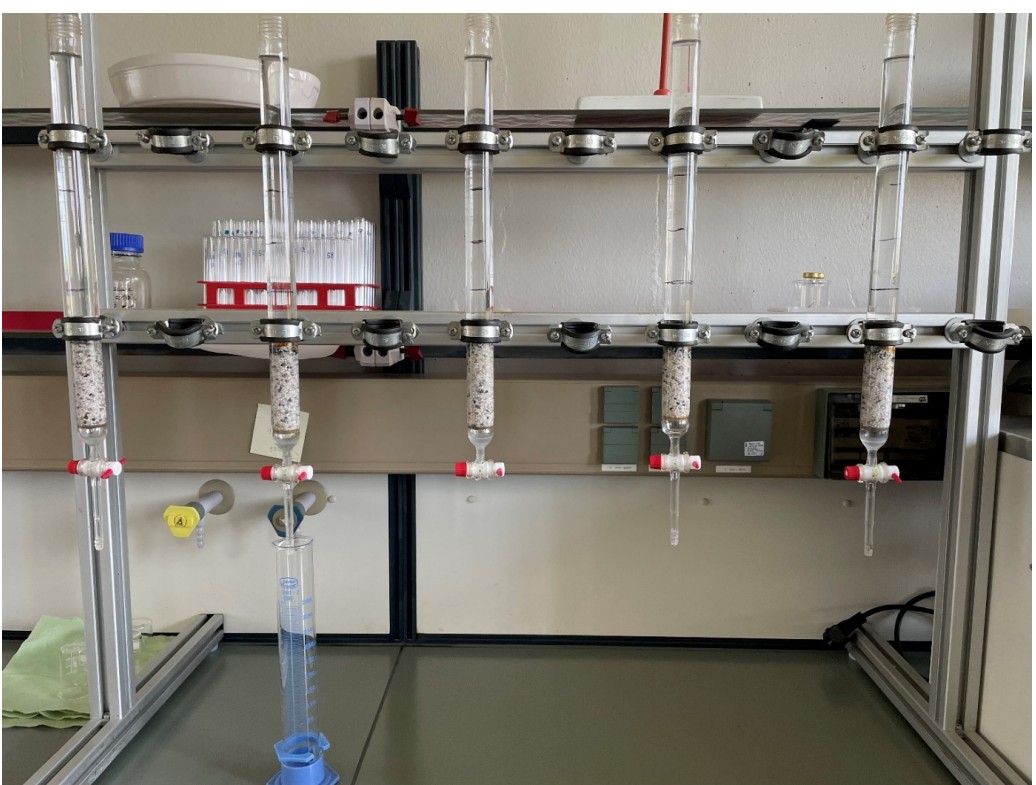

**Figure 1.** Column experimental set-up for $Fe^0$ leaching by ascorbic acid (2 mM). The photograph was made at the end of the experiments. The spout of the third column was broken during the experiments but this has no incidence on the performance of the system.

*2.5. Experimental Results*

Given that iron dissolution of $Fe^0$ and of iron corrosion products (FeCPs) is initially a linear function of the time [38] for a given time ($t_1 > t_0$) after the start of the experiment ($t_0 = 0$), the total iron concentration at $t_1$ ($[Fe]_t$) is a linear function as defined in Equation (2).

$$[Fe]_t = a \times t + b \qquad (2)$$

where a is the slope of the line, t represents the elapsed time since the immersion of $Fe^0$ in the leaching solution, and b the value of [Fe] at $t_0$. Ideally, b approaches 0.

The purpose of this study was to determine the timeframe for the linearity of Equation (2). The regression coefficients 'a' and 'b' are characteristic of each individual $Fe^0$ sample. In fact, 'a' represents the rate of dissolution of Fe from $Fe^0$ while 'b' gives an estimate of the amount of FeCPs or the fraction thereof that is dissolved by the used complexing agent (e.g., AA, EDTA and Phen). Note that AA and Phen form stable $Fe^{II}$ complexes, whereas EDTA forms stable $Fe^{III}$ complexes [42]. Accordingly, lower b-values are expected in AA tests. The a ($mg\ L^{-1}\ h^{-1}$) and b ($mg\ L^{-1}$) values derived from Equation (2) are converted to $\mu g\ h^{-1}$ and $\mu g$, respectively. Dissolution rates (a values equal to $k_{AA}$, $k_{EDTA}$, and $k_{Phen}$) were calculated using Origin software (Version 8.0).

**3. Results and Discussion**

*3.1. Suitability of the Experimental Protocol*

Figure 2a compares the extent of iron dissolution of ZVI1 in 2 mM AA, EDTA, and Phen. It can be seen that EDTA dissolves more $Fe^0$ than AA and Phen. The increasing order of performance (i.e., Fe leaching efficiency) is EDTA > AA > Phen. It should be recalled that: (i) EDTA forms very strong complexes with $Fe^{III}$ [59,60], (ii) AA reduces $Fe^{III}$ to $Fe^{II}$ and forms very stable complexes with $Fe^{II}$ [53], and (iii) Phen also forms very stable

complexes with $Fe^{II}$ but does not reduce $Fe^{III}$ [42]. Thus, the data indicate that EDTA $Fe^0$ produces by oxidative dissolution $Fe^{II}$ (Equation (1)), which is further oxidized to $Fe^{III}$ by $O_2$ in water (Equation (3)). In addition, some corrosion products are dissolved and stabilized in the aqueous phase [61] (Equation (4)). In other words, EDTA supports both the oxidative dissolution of $Fe^0$ and the dissolution of iron corrosion products. On the contrary, since $Fe^{II}$ AA complexes are very stable even in the presence of $O_2$, AA induces the oxidative dissolution of $Fe^0$ and the reductive dissolution of iron corrosion products. Finally, since Phen has no reductive power for iron corrosion products, dissolved Fe in its presence only results from oxidative dissolution of $Fe^0$. This reasoning fully justifies the observed order of leaching efficiencies (EDTA > AA > Phen). The corresponding a-values are: $k_{EDTA} = 18.6\ \mu g\ h^{-1}$; $k_{AA} = 13.2\ \mu g\ h^{-1}$; and $k_{Phen} = 8.1\ \mu g\ h^{-1}$ (Table 2).

$$4Fe^{2+} + O_2 + 2H^+ \Rightarrow 4Fe^{3+} + 2OH^- \tag{3}$$

$$FeOOH + EDTA + 3H^+ \Rightarrow [FeEDTA]^{3+} + 2H_2O \tag{4}$$

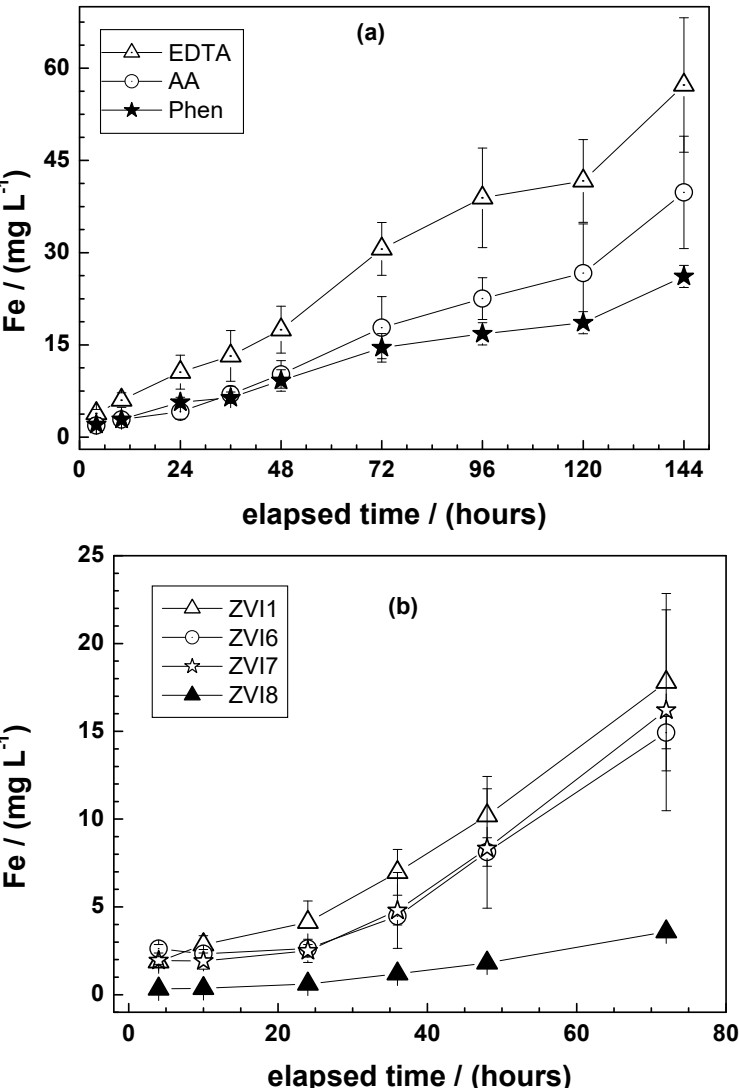

**Figure 2.** Time-dependent dissolution of $Fe^0$: (**a**) ZVI1 in 2 mM AA, EDTA or Phen using, and (**b**) ZVI1, ZVI6, ZVI7 and ZVI8 in 2 mM AA. Experimental conditions: V = 50 mL, $m_{ZVI}$ = 0.1 g, T = 25 ± 4 °C. The lines are not fitting functions, rather, they simply connect points to facilitate visualization.

**Table 2.** Statistical parameters (a = $k_{ligand}$, b, $R^2$) for the eight tested $Fe^0$ materials. Experimental conditions: initial ligand concentration 2 mM, room temperature $25 \pm 4\ °C$, and $Fe^0$ mass loading $2\ g\ L^{-1}$. The number of experimental data points are: 2 for b, 7 for a, 9 for $R^2$.

| Sample | b | $\Delta$(b) | a | $\Delta a$ | $R^2_{(7)}$ | $R^2_{(9)}$ |
|---|---|---|---|---|---|---|
| | ($\mu g$) | ($\mu g$) | ($\mu g\ h^{-1}$) | ($\mu g\ h^{-1}$) | (-) | (-) |
| **Using AA** | | | | | | |
| ZVI1 | 110.3 | 11.9 | 13.2 | 0.5 | 0.99 | 0.95 |
| ZVI2 | 108.5 | 10.4 | 17.2 | 1.1 | 0.98 | 0.92 |
| ZVI3 | 92.5 | 9.8 | 11.5 | 1.3 | 0.94 | 0.88 |
| ZVI4 | 118. 9 | 16.1 | 14.8 | 0.5 | 0.99 | 0.98 |
| ZVI5 | 119.4 | 15.6 | 12.3 | 0.7 | 0.99 | 0.94 |
| ZVI6 | 126.1 | 8.1 | 10.3 | 0.8 | 0.97 | 0.78 |
| ZVI7 | 96.9 | 1.0 | 13.4 | 0.6 | 0.99 | 0.57 |
| ZVI8 | 16.9 | 2.1 | 2.8 | 0.1 | 0.99 | 0.90 |
| **ZVI1 using AA, EDTA, and Phen** | | | | | | |
| AA | 110.3 | 11.9 | 13.2 | 0.5 | 0.99 | 0.95 |
| EDTA | 56.3 | 76.7 | 18.6 | 1.3 | 0.98 | 0.99 |
| Phen | 66.4 | 35.1 | 8.1 | 0.5 | 0.98 | 0.99 |

Previous efforts to characterize the intrinsic reactivity of $Fe^0$ using ligands (e.g., EDTA, Phen) [34,38,41,42] could not explain a negative b-value from Equation (2). It has been postulated that b represents the amount of Fe leached from iron corrosion products, thus, b should be necessarily greater than or equal to zero. Because AA reduces (some) iron corrosion products before stabilizing them as $Fe^{II}$-AA complexes, the AA method is expected to solve this open issue of negative b values.

### 3.2. Deciphering the Processes of Iron Dissolution in $Fe^0$/AA Systems

Figure 2b compares the time dependent iron dissolution of 4 selected ZVIs in 2 mM AA. As a rule, the more reactive a material is under given conditions, the greater the $k_{AA}$ value. It can be seen that ZVI1 clearly exhibits a higher dissolution rate, while ZVI6 and ZVI7 are very close. ZVI8 shows the slowest iron dissolution. The dissolution of different $Fe^0$ materials in AA (2 mM) is one goal of this research and will be discussed in the next section for the 8 materials tested (Figure 3). This section focuses on the initial phase of iron dissolution (first 72 h) for four selected materials as representatives for different reactivities [42,43].

As discussed in Section 2.5, dissolved Fe ($Fe^{II}$-AA) results from two concurrent processes: (i) oxidative dissolution of $Fe^0$, and (ii) reductive dissolution of FeCPs. In other words, there is competition for AA, but $Fe^0$ is present in large excess [33,34,41] and FeCPs are poorly crystalline in structure [62–66] and thus comparatively readily soluble [67–71]. Accordingly, it can be considered that the reductive dissolution of FeCPs is quantitative in the early phase of the experiments. Operationally, it is considered herein that Fe is quantitatively extracted from FeCPs by 2 mM AA. This approach is borrowed from the sequential extraction methods of soil chemical analysis [72–75], acknowledging difficulties in reducing iron oxides, even under acidic conditions and in the presence of chelating agents [73]. Clearly, it is considered, that the amount of Fe extractable from FeCPs is dissolved in the early phase of the experiment.

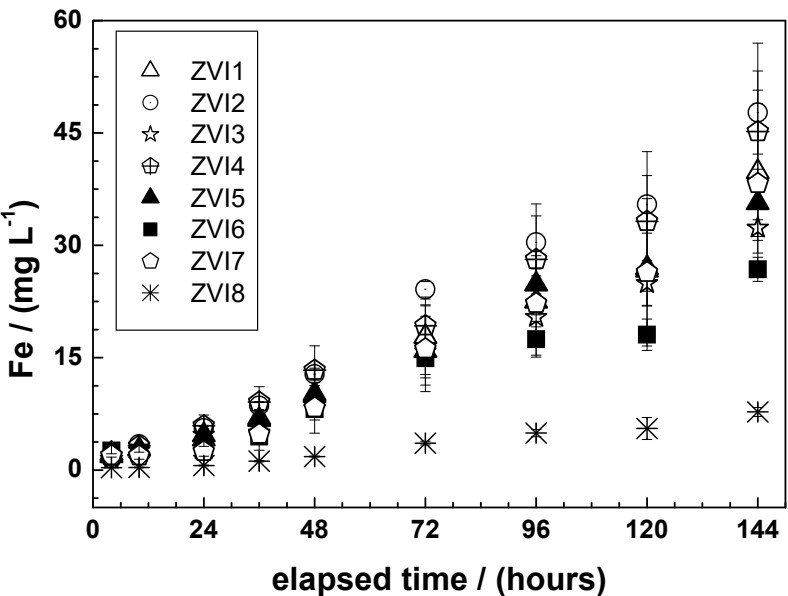

**Figure 3.** Time-dependent concentrations of dissolved Fe of from the 8 $Fe^0$ samples in the presence of 2 mM AA for the 8 ZVIs tested in this works. For $k_{AA}$ values (Table 2), only data for t ≥ 24 h are considered. Experimental conditions: V = 50 mL, $m_{ZVI}$ = 0.1 g, T = 25 ± 4 °C, $Fe^0$ mass loading 2 g $L^{-1}$.

The main feature evident in Figure 2b is that Fe dissolution is very slow during the first day of the experiment (t < 24 h) and then increases progressively. This observation suggests that, under the experimental conditions used in the present study (e.g., 0.1 g $Fe^0$, 2 mM AA), the fraction of Fe resulting from reductive dissolution of iron corrosion products is relatively small. Except for ZVI1, the other three materials did not experience any significant Fe leaching before t = 24 h. For ZVI8, the dissolution rate remains very low until the end of the experiments. Therefore, it can be assumed that the b value corresponds to the level of iron dissolution after about 10 h. In this study, the value obtained for 6 h is tabulated (Table 2) and the a-value or $k_{AA}$ corresponds to the slope of the line [Fe] = f(t) for t ≥ 24 h. In other words, when characterizing the $Fe^0$ intrinsic reactivity using the AA method, the Fe concentration [Fe] after some 4 to 10 h is used to determine the b value (e.g., in μg) while the [Fe] values for t ≥ 24 h are used to determine a or $k_{AA}$ (e.g., μg $d^{-1}$). This rule is used in the next section.

### 3.3. Characterizing $Fe^0$ Dissolution in 2 mM Ascorbic Acid (AA)

Figure 3 compares the extent of iron dissolution of all 8 tested ZVIs in 2 mM AA over 144 h (6 days). It can be seen that the intensity of $Fe^0$ dissolution increases slowly from day 1 (24 h) to the end of the experiment. The corresponding dissolution rates (a values from Equation (2) or $k_{AA}$ values) are summarized in Table 2. As explained in Section 3.2, the first two data points (t < 24 h) are applied to calculate the b values, while the remaining 7 points (t ≥ 24 h) are used to calculate the a or $k_{AA}$ values. Table 2 also shows the coefficient of determination $R^2$ for all 9 experimental data points for all ZVIs. $R^2$ is the proportion of the variation in the dependent variable that is predictable from the independent variable. The poorer regression and the associated negative values of b (e.g., μg) justify the use of the data obtained for 4 and 10 h to calculate the b values as shown in Tabel 2. At the same time, this suggests a simplification of the experimental procedure as there is no need for more than one sampling event per day. On the other hand, the last experimental data point can be ignored so that the protocol for $Fe^0$ characterization using the AA method is reduced to one working week of five days. The suggested sampling times during these five days are: 6, 24, 48, 72, 96, and 120 h. The b-values in Table 2 (17 to 127 μg) suggest that less 1% of the different $Fe^0$ materials are iron corrosion products.

A key feature of the AA test is that apart from the two experimental points at t < 24 h, all other points are used as they obey the 'linearity' request. Accordingly, all 7 points were used to determine $k_{AA}$. In the EDTA test on the contrary, only some few points (e.g., four or five) were useful for the $K_{EDTA}$ value of readily reactive materials (e.g., ZVI1). In fact, considering experimental points corresponding to longer duration yielded to poorer $R^2$ values. The findings of this study clearly demonstrated that this 'disturbance' is attributed to the presence of air oxygen (and the stability of $Fe^{III}$-EDTA complexes) [59,60].

Based on Figure 3, ZVI8 is by far the least reactive material of the eight materials tested. Based on the $k_{AA}$ values in Table 2, the reactivity decreases in the following order:

ZVI2 > ZVI4 > ZVI7 > ZVI1 > ZVI5 > ZVI3 > ZVI6 >> ZVI8.

This order of reactivity corresponds to that determined for the same $Fe^0$ materials by the EDTA method [34], the Phen method [41,42] and the $H_2$ method [43]. However, more data are lacking to allow a detailed discussion because the methods involved have not been independently tested or used. Although the EDTA method is already two decades old [76,77], it has never been tested by other research groups. However, this method was not published in a peer-reviewed journal until 2005 [38]. Nearly 18 years passed and several other independent methods have been presented [39,40,68], but are still not universally tested and accepted [21,35,78]. It is very disappointing that the $Fe^0$ research community has been working for 30 years without characterizing the $Fe^0$ materials that are at the center of their systems. Considering the eight materials tested here, the $k_{AA}$ values vary from 17.2 for ZVI2 to 2.8 for ZVI8, giving a reactivity ratio of more than 6. This, means that six times more $H_2$ or $Fe^{2+}$ is produced in the ZVI2 system than in the ZVI8 system under similar operating conditions. Clearly, one researcher testing ZVI8 may conclude that $Fe^0$ is not suitable, while a colleague testing ZVI1 (under similar conditions) may strongly recommend $Fe^0$ for the same application.

As mentioned above, a deeper discussion is not possible due to the lack of comparable approaches. Unfortunately, research on $Fe^0$ reactivity has mostly been a race for the most reactive material (i.e., bimetallic, nano-$Fe^0$) [22,78]. What is needed, however, are appropriate materials specific for the problems on site. For example, ZVI8, the least reactive material can be the best material for a field situation where aggressive environmental conditions (such as acid mine drainage) sustain iron corrosion in the long term. In such situations, a more reactive material (e.g., ZVI2) would lead to clogging of the system, or at least result in material wastage, since a larger proportion of the corroded $Fe^0$ is not serving the remediation goal. The remainder of this paper discusses the long-term kinetics of $Fe^0$ dissolution (corrosion rate).

*3.4. Characterizing the Long-Term $Fe^0$ Dissolution in Column Studies*

Figure 4 and Table 3 compare the extent of iron dissolution behaviour of three selected ZVIs (ZVI1, ZVI3 and ZVI5) in 2 mM AA in column experiments for 55 leaching events. Figure 4a shows that 2 to 12 mg of Fe could be leached daily from each column containing 1.0 g of $Fe^0$. Figure 4b shows that up to 530 mg of Fe could be leached after 55 leaching within 129 days. The $Fe^0$ reactivity increases in the order ZVI1 < ZVI5 < ZVI3. The high reactivity of ZVI3 is due to its higher porosity and surface area compared to the other materials. The same order of reactivity was reported in related works [34,43].

For each material, the amount leached was high at the beginning of the experiment, and then progressively decreased with increasing leaching events (elapsed time) until about 70 days (Figure 4a). After the decrease of Fe concentrations between about 70 and 105 days, they increased again to values comparable to the initial values (Table 4). The trend was the same for all $Fe^0$ specimens with relatively little variations between the samples. The cumulative extent of Fe leaching shows that ZVI3 had a slightly higher Fe leaching efficiency over the 55 leaching events than the other two samples. Taken together, Figure 4a,b illustrates clear material-specific trends in the long-term kinetics of iron corrosion that are well known to iron corrosion scientists [79–82] but has not been really addressed within the $Fe^0$ remediation research community [83–85].

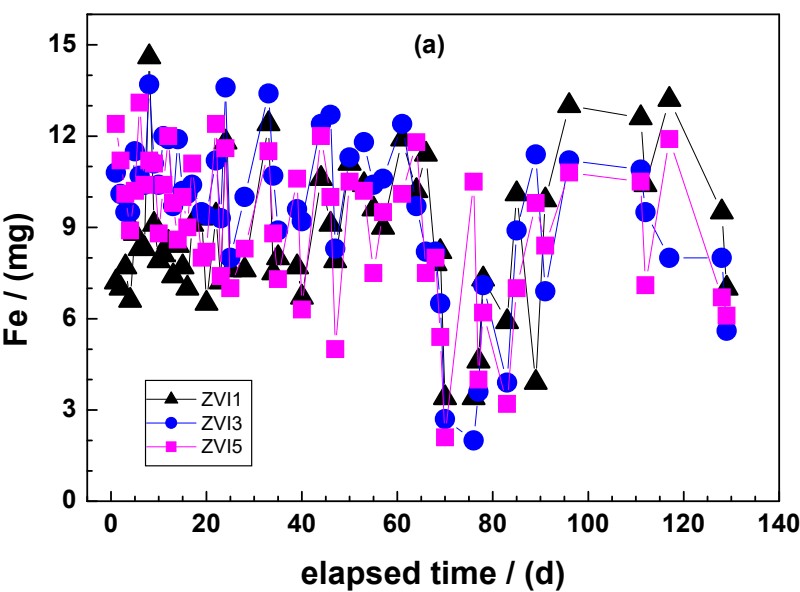

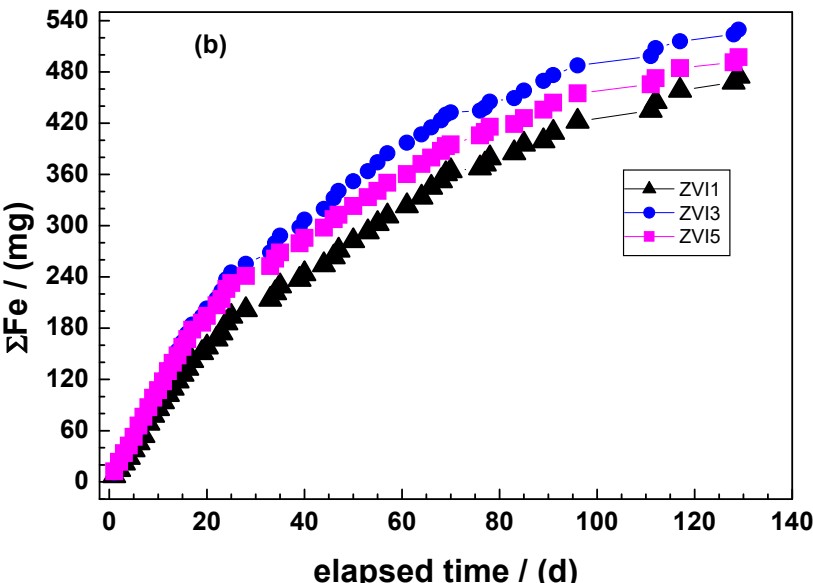

**Figure 4.** Time-dependent extent of Fe leaching from the three $Fe^0$ specimens tested: (**a**) mass (mg) released per leaching event, and (**b**) cumulative mass. Experimental conditions: $m_{iron}$ = 1.0 g; [AA] = 2 mM; and T = 25 ± 4 °C. The lines are not fitting functions. They simply connect points for ease of visualization.

**Table 3.** Summary of the amount of Fe leached from the $Fe^0$ specimens tested over 55 leaching events. The average daily leached amount is the sum of the leached mass divided by 129. Experimental conditions: $m_{iron}$ = 1.0 g; [AA] = 2 mM; and T = 25 ± 4 °C.

| Rate (Unit) | | ZVI1 | ZVI3 | ZVI5 |
|---|---|---|---|---|
| Daily | (mg) | 3.7 | 4.1 | 3.9 |
| Total | (mg) | 475 | 530 | 497 |
| Total | (%) | 47.5 | 53.0 | 49.7 |

**Table 4.** Event-specific Fe leached mass (mg) from 1.0 g of the tested Fe$^0$ specimens at 8 selected events. The Fe$^0$ specimens are ordered from left to right in the order of increasing value of the mass after the second leaching event, corresponding to day 2 of the experiment.

| Event | Time | ZVI1 | ZVI3 | ZVI5 |
|---|---|---|---|---|
| (-) | (d) | (mg) | (mg) | (mg) |
| 2 | 2 | 7.0 | 10.1 | 11.2 |
| 10 | 10 | 7.9 | 10.4 | 8.8 |
| 20 | 22 | 9.4 | 11.2 | 12.4 |
| 30 | 44 | 10.6 | 12.4 | 12.0 |
| 40 | 68 | 7.8 | 8.2 | 8.0 |
| 50 | 96 | 13.0 | 11.2 | 10.8 |
| 51 | 111 | 12.6 | 10.9 | 10.5 |
| 52 | 112 | 10.4 | 9.5 | 7.1 |

A combination of (i) non-constant kinetics of iron corrosion for individual materials, and (ii) different laws of the variation kinetics amount materials make any prediction of the leaching extent challenging (Table 4). Table 4 shows that for the first 10 leaching events, the increasing order of reactivity was ZVI1 < ZVI3 < ZVI5. Between the 10th leaching event and the 52th there is no uniform trend in the variation of the extent of Fe leaching from the three tested materials. Summarized, these results of long-term Fe$^0$ leaching using AA have confirmed that using constant value of corrosion rates in modelling efforts cannot be supported by any assumption [86–88].

## 4. Significance of the Results

### 4.1. Fe$^0$ Quality as a Stand-Alone Operational Parameter

It is intuitive that different Fe$^0$ materials would provide different results of water treatment under given operating conditions. This is because each Fe$^0$ sample is unique in its intrinsic reactivity [26,27,38,89,90]. Despite this evidence, little attention has been paid to the Fe$^0$ source (Fe$^0$ quality or Fe$^0$ type) as a stand-alone operational variable for the efficiency of Fe$^0$/H$_2$O systems [21,35,39,40,78]. For example, Westerhoff [26] tested six different Fe$^0$ specimens for nitrate removal at an initial pH of 2.0. His results clearly showed that the kinetics and the extent of NO$_3^-$ removal and pH change varied as a function of the Fe$^0$ source (Fe$^0$ quality), ranging from 20% to 100% after 4 h. For further investigations, the author selected a Fe$^0$ specimen with intermediate efficiency ("partial, but not complete NO$_3^-$ removal") in order to study other parameters such as the nature and amount of reaction products. In retrospect, it can be said that such pragmatic approaches have generally been used within the Fe$^0$ research community to select tested Fe$^0$ materials. In support of this statement, while various Fe$^0$ samples have been used in Fe$^0$-based subsurface permeable reactive barriers (PRBs), available models for Fe$^0$ PRBs use a corrosion rate of a single Fe$^0$ sample based on H$_2$ evolution [36,37], although Reardon [36] reported eight different corrosion rates for various Fe$^0$ specimens [85,87]. Obviously, the selection of Fe$^0$ for field applications has not been justified by quality assurance and quality control.

### 4.2. Other Key Operational Parameters

The performance of Fe$^0$ applications in the field has been shown to depend on: (i) the acidity of the influent (pH value), (ii) the redox conditions (Eh value), (iii) the concentrations of co-solutes (e.g., Ca$^{2+}$, Cl$^-$, Mg$^{2+}$, NO$_3^-$, HPO$_4^{2-}$, SO$_4^{2-}$), and (iv) the Fe$^0$ dosage or Fe$^0$ quantity [4,6,91–94]. The relative importance of these parameters in determining the performance of Fe0 PRBs has been established since 2007 [1,7,24]. In 2007, it was timely noted that there is a lack of field data to address the long-term performance of Fe$^0$ PRBs in terms of reactivity and permeability. Unfortunately, 15 years later, the results confirming the efficiency of Fe$^0$ PRBs [91,95–97] do not address the questions in terms of filling any knowledge gaps in the past [7,10,17,24]. In other words, the intrinsic reactivity of Fe$^0$ (Fe$^0$ quality not Fe$^0$ quantity) has yet to be considered as a key stand-alone variable in the

design of $Fe^0/H_2O$ systems for remediation [35,39,40,42,87]. Factors affecting the intrinsic reactivity of $Fe^0$ include its porosity, shape, size and size distribution, surface layers, surface area, and surface smoothness [78]. Their importance as operational parameters have been demonstrated in various studies [34,35,38,92,96], but not to the extent that sound science-based recommendations can be made to engineers seeking for appropriate $Fe^0$ specimens for site-specific applications [43,98].

*4.3. Current Approaches to Address $Fe^0$ Quality*

The $Fe^0/H_2O$ interface is a heterogeneous system in which interacting reactions are of central importance [1,10,22,99]. Contaminant removal in $Fe^0/H_2O$ systems is largely controlled by processes occurring at two interfaces: (i) $Fe^0$/oxides and oxides/$H_2O$; and (ii) within the oxide scale. Accordingly, various spectroscopic/microscopic techniques have been used in an effort to gain more detailed knowledge enabling to understand the processes that influence the long-term efficiency of $Fe^0/H_2O$ systems [78,100,101]. For example, modern analytical techniques such as Mössbauer spectroscopy, Raman spectroscopy, scanning electron microscopy (SEM), X-ray absorption spectroscopy (XAS), X-ray diffraction (XRD), and X-ray photoelectron spectroscopy (XPS), have been used complementarily to characterize samples from laboratory and field investigations [3,78,102]. However, by characterizing the nature or amount of uncorroded $Fe^0$, the generated corrosion products, and the speciation of contaminants at selected times (e.g., at the end of the experiment), these tools provide only a snapshot of the system at those selected times [103–106]. Moreover, the duration of the experiments is typically too short (e.g., a few days or weeks) to be representative for of water filters and reactive barriers that are expected to operate for years or decades [1–4]. Clearly, the conventional approach does not account for the documented non-linear decrease of the corrosion rate which is critical to the design of sustainable remediation systems (Section 3). In other words, despite three decades of intensive research in water remediation, there is still little guidance on how to relate the intrinsic properties of a $Fe^0$ sample ($Fe^0$ quality) to its observed long-term field performance. Only eight peer-reviewed publications were found that focus on characterizing the intrinsic reactivity of $Fe^0$ materials (Table 5) with a perspective of introducing a standard protocol.

**Table 5.** Summary information of the peer-reviewed publications on the characterization of the intrinsic reactivity of Fe0 materials and their citation frequency according to Google Scholar until 2023 (accessed on 18 January 2023).

| Anno | Title | Citations (Total) | Citations (per Year) |
|---|---|---|---|
| 1995 | Anaerobic corrosion of granular iron: Measurement and interpretation of hydrogen evolution rates | 386 | 13.8 |
| 2005 | Testing the suitability of zerovalent iron materials for reactive walls | 110 | 6.1 |
| 2014 | Standardization of the reducing power of zerovalent iron using iodine | 30 | 3.3 |
| 2015 | Simple colorimetric assay for dehalogenation reactivity of nanoscale zero-valent iron using 4-chlorophenol | 35 | 4.4 |
| 2016 | A facile method for determining the Fe(0) content and reactivity of zero valent iron | 47 | 6.7 |

**Table 5.** *Cont.*

| Anno | Title | Citations (Total) | Citations (per Year) |
|------|-------|-------------------|----------------------|
| 2019 | A novel and facile method to characterize the suitability of metallic iron for water treatment | 37 | 9.3 |
| 2020 | Characterizing the reactivity of metallic iron for water treatment: $H_2$ evolution in $H_2SO_4$ and uranium removal efficiency | 8 | 2.7 |
| 2020 | Cost-effective remediation using microscale ZVI: comparison of commercially available products | 6 | 2.0 |

For the sake of completeness, it should be acknowledged that Fisher and Feinberg [98] recently introduced a new approach to characterize the extent of $Fe^0$ consumption in $Fe^0/H_2O$ systems. This innovative approach roots on the differential magnetic susceptibility between $Fe^0$ (and magnetite: $Fe_3O_4$) and Fe minerals such as goethite (FeOOH) and hematite ($Fe_2O_3$). In fact, a decrease in magnetic susceptibility tracks the conversion of high susceptibility materials ($Fe^0$ and $Fe_3O_4$) to lower susceptibility minerals (FeOOH and $Fe_2O_3$). Accordingly, if one measures the initial magnetic susceptibility of newly installed $Fe^0$ filter, continued measurements will indicate the remaining capacity of the media to provide additional corrodible iron. Remember that, in water remediation $Fe^0$ is converted to high-surface-area iron oxide minerals (FeCPs) that are excellent contaminant scavengers.

The methods in Table 5 are based either on (i) monitoring the formation of primary iron corrosion products ($Fe^{II}$ and $H_2$) [36,38,42,43] or (ii) using some easy-to-monitor reactions with some reactive species [35,39,40,106]. However, they have not been routinely used for quality assurance and quality control (QA/QC), and they are not very user friendly. For example, the very first test by Reardon [36] is 28 years old and has been cited only 386 times, an average of 14 times per year. This is practically insignificant in a context where hundreds of articles on $Fe^0$ remediation are published every year [107–111]. The Reardon test [36] requires large quantities of $Fe^0$ (about 500 g) and sophisticated equipment to monitor the $H_2$ evolution [34,42,90,110]. The Phen test using 1,10-Phenanthroline to complex $Fe^{II}$ from iron corrosion suffers from the toxicity of this chemical. Therefore, there is still need for safe, affordable and applicable methods for QA/QC of $Fe^0$. The AA method is presented here as a candidate method for routine QA/QC.

### 4.4. The AA Method as a Quality Control Tool for $Fe^0$ Materials

We have developed a simple tool to characterize the intrinsic reactivity of commercially available granular metallic iron materials ($gFe^0$) by measuring the iron content in a dilute ascorbic acid solution (2 mM) within one week (5 or 6 days). Ascorbic acid (vitamin C) is inexpensive, nontoxic, and readily available worldwide.

The protocol of the AA test can be summarized as follows:

(1) Add 0.1 g of $Fe^0$ to 50 mL of a 2 mM AA solution and monitor the concentration of dissolved Fe for 0.3, 1.0, 2.0., 3.0, 4.0 and 5.0 days.

(2) Use the iron concentration after 8 h to estimate the amount of iron corrosion products and the remaining data to determine the $k_{AA}$ value. $k_{AA}$ is the slope of the line dissolved [Fe] versus time t for t $\geq$ 24 h.

In future work, each $Fe^0$ material should be routinely presented in the experimental section with its $k_{AA}$ value. Such an approach will be similar to how conventional filter materials such as activated carbons are presented with their iodine number, removal capacity for methylene blue, or specific surface areas (SSA) [112–116]. Within a reasonable period of time, e.g., two years, it will be possible to better discuss the suitability of these

materials in terms of the parameters reported by the manufacturers (e.g., iron content, SSA). The task is to validate a simple test for quality assurance and quality control of $Fe^0$ materials. The AA method is valuable for users to effectively optimize designs of remediation systems, from setting technological parameters to budgeting (life cycle analysis LCA). The AA method is also expected to support the development of new $Fe^0$ materials with tailored properties for specific applications, such as "$Fe^0$ for saline wastewaters", "$Fe^0$ for acid mine drainage", or "$Fe^0$ for carbonate-rich waters". This approach will certainly support the design of $Fe^0$ filters for decentralized safe drinking water provision [83,87,107–121] and keep the international community on track to achieve Goal 6 ('Ensure access to water and sanitation for all') of the Sustainable Development Goals (SDGs) of the United Nations even for poor and vulnerable populations [122–125].

Activated carbons for specific applications can be developed because a new material is considered fully characterized when the following is specified: (i) carbon content (%), (ii) specific surface area ($m^2 \cdot g^{-1}$), (iii) pore volume ($cm^3 \cdot g^{-1}$), (iv) pore size distribution (e.g., percent of micropores), and (v) surface functional groups [112–116]. Once these are known, it suffices to roughly consider solubility and molecular size of the pollutant, as well as solution pH and the presence of other species (e.g., co-solutes) to select an appropriate activated carbon for a specific application. Attia et al. [115] summarized the following rule of thumb for selecting activated carbons: "The most widely used activated carbons are microporous and have high surface areas, and as a consequence, show high efficiency for the adsorption of low molecular weight compounds and low for larger molecules. The adsorption of bigger size compounds such as dyes, dextrines or natural organic compounds, requires materials with high mesopore contribution to the total pore volume of adsorbents". The present work has starting paving the way for such a rule of thumb for the characterization of $Fe^0$ materials for water remediation.

## 5. Conclusions

The efficiency of $Fe^0$ materials for environmental remediation and water treatment is certainly related to the oxidative dissolution of used samples. In characterizing $Fe^0$ leaching in 2 mM ascorbic acid, this study has demonstrated the complexity of intrinsic reactivity as a stand-alone operational parameter that must be carefully considered in the further development of an already established technology. The results suggest that materials that are efficient in short term laboratory experiments may not continue to react uniformly over the life time of the system. Although some of the observed differences could be explained by some known properties of the $Fe^0$ material (e.g., porosity of ZVI2), it should be explicitly stated that the discussion of material-related properties (e.g., Fe content, particle shape and size) is beyond the scope of this study. The discussion is based solely on the observed extent of iron dissolution observed under laboratory conditions. The situation may change once the $Fe^0$ is placed in the subsurface (e.g., carbonate rich, anoxic, saline environments).

The AA approach presented here is an improved version of a 20-year-old method using EDTA as a complexing agent to specify the oxidative dissolution of $Fe^0$. Systematic research is needed to further develop the AA method into a unified standard protocol for quality control/quality assessment of $Fe^0$ materials. Systematic testing of micro-sized and nano-sized $Fe^0$ as well as Fe alloys and sulfidized counterparts is required. The goal is to create a database of $Fe^0$ materials to select the right Fe product for site-specific applications.

**Author Contributions:** X.C., R.T., M.X. and C.N. conceived the presented idea and developed the theory. X.C., R.T. and M.X. carried out the experiment. R.H., H.R., W.G. and C.N. supervised this work. C.N. supervised the redaction of the first draft by X.C., R.T. and M.X. All authors discussed the results and contributed to the final manuscript. All authors have read and agreed to the published version of the manuscript.

**Funding:** X.C. is supported by a research grant from the China Scholarship Council (Project Code: 202006710005).

**Data Availability Statement:** Data are available on request.

**Acknowledgments:** The manuscript was improved thanks to the insightful comments of anonymous reviewers from Water. We acknowledge support by the German Research Foundation and the Open Access Publication Funds of the Göttingen University.

**Conflicts of Interest:** The authors declare no conflict of interest.

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
