# Peer review of "Developing the Ascorbic Acid Test: A Candidate Standard Tool for Characterizing the Intrinsic Reactivity of Metallic Iron for Water Remediation"

_water, doi:10.3390/w15101930_

Round 1
Reviewer 1 Report
The authors elaborately presented the results of a novel protocol to determine the quality of gFe0 based on ascorbic acid test. The results of the reported protocol were compared with those of widely applicable methods such as EDTA and Phen. The manuscript represents a systematic yet remarkable work to validate an assay technique in the area of analytical chemistry. However, some issues are raised as follow:
1. The effectiveness of Phen method for determining the intrinsic reactivity of Fe0 is relied on the fact that it addressed the forward dissolution of Fe0 with neglecting the contribution from dissolving corrosion-based species. In the present work, AA induces both the oxidative dissolution of Fe0 and the reductive dissolution of atmospheric corrosion products. Therefore, I think AA test will show an overestimation for the reactivity of Fe0 specimens (did not handle the limitation of EDTA method).
2. I wonder why the batch experiments to assign the time-dependent dissolution of Fe0 in AA, Phen and EDTA were not continued for enough time as in previous work (Water 2019, 11, 246).
Author Response
Comments 1: The effectiveness of Phen method for determining the intrinsic reactivity of Fe0 is relied on the fact that it addressed the forward dissolution of Fe0 with neglecting the contribution from dissolving corrosion-based species. In the present work, AA induces both the oxidative dissolution of Fe0 and the reductive dissolution of atmospheric corrosion products. Therefore, I think AA test will show an overestimation for the reactivity of Fe0 specimens (did not handle the limitation of EDTA method).
Responses 1: Many thanks for this remark. The key is that the amount of atmospheric corrosion products is limited and its reductive dissolution is rapid. Thus, discarding the data for t < 24 h has successfully addressed this issue (Table 2). We have now considered this "kinetic argument" in the discussion.
Comments 2: I wonder why the batch experiments to assign the time-dependent dissolution of Fe0 in AA, Phen and EDTA were not continued for enough time as in previous work (Water 2019, 11, 2465).
Responses 2: Many thanks for this important remark. All three methods rely on the time frame of the linearity of [Fe] = f(t) (Fe0 oxidative dissolution). For AA (this work) the linearity was observed prompt after the disturbance created by the dissolution of atmospheric corrosion products. In other words, while the EDTA method has to sort out some experimental points at longer contact time (e.g. t > 36 h), the AA method has ignored (2) points corresponding to lower contact times (e.g. t < 24 h) as depicted in Table 2.
Noubactep, C.; Meinrath, G.; Dietrich, P.; Sauter, M.; Merkel, B.J. Testing the suitability of zerovalent iron materials for reactive walls. Environ. Chem. 2005, 2, 71–76.
Lufingo M., Ndé-Tchoupé A.I., Hu R., Njau K.N., Noubactep C. (2019): A novel and facile method to characterize the suitability of metallic iron for water treatment. Water 11, 2465.
Reviewer 2 Report
see the attached comments.

Author Response
Comments 1: The author should add another table of comparison of the present study with previous study to evaluate the effectiveness of AA test. Done, Thanks!
Comments 2: In line 218, the formula numbers should be aligned. Done, Thanks!
Comments 3: The format of Table 1, Table3, and Table 4 is inconsistent. Done, Thanks!
Comments 4: To improve your chances of success, authors are advised to improve the level of English throughout your manuscript. You may wish to ask a native speaker to check your manuscript for grammar, style and syntax. Done, Thanks!
Comments 5: In line 385: “The performance of field Fe0 applications has been demonstrated to depend on: (i) the acidity of the influent (pH value), (ii) the redox conditions (EH value), (iii) the concentrations of co-solutes (e.g. Ca2+, Cl−, Mg2+, NO3−, HPO42−, SO42−), and (iv) the Fe0 dosage or Fe0 quantity”. What is the impact of the parameters mentioned above on this experimental study (AA test).
Responses 5: Thanks for this suggestion. We have planned a stand-alone paper on the effects of co-solutes (mainly anions: Cl−, HCO3−, HPO42−, NO3−, SO42−) on the initial kinetics of Fe0 dissolution. The present study aims at insisting on Fe0 type (intrinsic reactivity) as stand-alone design parameter for remediation systems.
Reviewer 3 Report
The article "Developing the ascorbic acid test: A candidate standard tool for characterizing the intrinsic reactivity of metallic iron for water remediation" is devoted to the development of a simple spectrophotometric method for characterizing ZVI based on the degree of iron dissolution in ascorbic acid solution. The AA method will support the development of new Fe0 materials with desired properties, for example for saline wastewater, acid mine drainage or carbonate-rich waters.
I believe that the manuscript has a high scientific and practical significance and can be published in the journal Water after a minor revision. Some tips for improving the article are given below.
l. 28, 140, 152, 234, 255 ... eight (8) – there is no need to give an entry in both forms.
Table 1 State what is meant by size for each specific ZVI shape
l.159 Was the sample mixed during the experiment or before sampling?
l.160, 164 repetition of clarifications about the triple experiment
Fig. 1 On one of the columns you can imagine where sand, solution and Fe0 sample are located. What happened to the center column spout?
l.184 decipher the designation of FeCPs
l.217, 250, 251, 286 misplaced characters
Author Response
Some tips for improving the article are given below.
Line 28, 140, 152, 234, 255 ... eight (8) – there is no need to give an entry in both forms. Corrected, thanks!
Table 1 State what is meant by size for each specific ZVI shape. Done, thanks!
Line 159 Was the sample mixed during the experiment or before sampling? Quiescent experimental protocol (as specified in Line 162).
Line 160, 164 repetition of clarifications about the triple experiment.
Corrected, thanks!
Fig. 1 On one of the columns you can imagine where sand, solution and Fe0 sample are located. What happened to the center column spout?
The column spout was broken during the experiments but is has no incidence on the continuation. The mass of ZVI is just 1 g, it cannot been seen.
Line 184 decipher the designation of FeCPs. Done, thanks!
Line 217, 250, 251, 286 misplaced characters. Corrected, thanks!
Reviewer 4 Report
This manuscript reports the results of using a ascorbic acid test to evaluate the reactivity of metallic iron. Although some of the results described would need further study, my opinion is that the article could be acceptable after minor revision.
Please correct some misspelling mistakes (for instance: line 45, agent (without “s”), line 262, corresponding); also avoid using capital letter in the “o” to formulate o-phenanthroline; please revise and correct the arrow symbol in the equations 1-4; please correct the format of the different paragraphs (leading, letter size, etc.); please write the sentence “protons and only protons” (line 48) in a more understable way. English language and style are fine/minor spell check required
References 71 and 74 are repeated.
Author Response
Please correct some misspelling mistakes (for instance: line 45, agent (without “s”), line 262, corresponding); also avoid using capital letter in the “o” to formulate o-phenanthroline; please revise and correct the arrow symbol in the equations 1-4; please correct the format of the different paragraphs (leading, letter size, etc.); please write the sentence “protons and only protons” (line 48) in a more understable way. English language and style are fine/minor spell check required
Many thanks, we have fixed all these points. Thanks particularly for the correction on "ortho". This is a kind of mistake that should not be introduced in the scientific literature. Actually it is easy to oversee. Thanks!
References 71 and 74 are repeated. Fixed, Thanks!
Reviewer 5 Report
The article has an unusual structure. It is not only a research article but also a review. The authors showed that data obtained in laboratory experiments do not necessarily reflect real problems. In addition to the research results (to which I have no substantive comments), the authors refer to other publications on the subject, making not only a standard literature review (Intruduction), but also giving the work a review character supported by their own experiment.
This approach required a significant amount of work. I believe that the content of the work is at a high level. Before publishing, you should think about the editing side. Some paragraphs have different spacing and indentation. In addition, there are numerous editing errors in the work, such as "40%" instead of "40%". In addition, it suggests that the authors group the figures, eg Fig. 2 and Fig. 3 together would be more readable. I also believe that Authors should limit the number of self-citations. The authors cite about 20 of their own works out of 92 items, some of which are grouped.
Author Response
Before publishing, you should think about the editing side. Some paragraphs have different spacing and indentation. In addition, there are numerous editing errors in the work, such as "40%" instead of "40%".
We have carefully considered all these points, thanks!
In addition, it suggests that the authors group the figures, eg Fig. 2 and Fig. 3 together would be more readable.
Done thanks, it is now Figure 2 a and 2b.
I also believe that Authors should limit the number of self-citations. The authors cite about 20 of their own works out of 92 items, some of which are grouped.
The reviewer is absolutely right. However, we have just selected our most relevant previous works, based on criteria like content and accessibility. For example the very original publication on thye EDTA method is not referenced (2004 - Conference). Noubactep‘s thesis (2003) is prefered as it is readily accessible. This remark is recurrent and it justified from our end, that we have went so many paths alone. It is not a pleasure for us to negatively referencing colleagues. Weh ad to do that in the eraly stage (2007 to 2014) but now that an important body of own work is available, an healthy mixture is found for each presentation. In essence, because we are standing walking (almost) alone and there are two schools of thought, 50% self-citation is acceptable. However, we are far below this level!
To illustrate this with the ZVI characterization, we have introduced the EDTA method in 2004 (peer-reviewed journal 2005), have revisited it in 2013, introduced the Phen method in 2019. No other research group has considered them and researchers introducing new methods have not considered them. We have tried at our best to consider other achievements in the best positive manner we could.
Round 2
Reviewer 1 Report
After the modifications, I think the authors do not reply to the previous points adequately.
Although the enclosed work investigated the suitability of AA method to assign the reactivity of Fe0 specimens, the revised manuscript contains some ambiguous statement, Fore example:
1. Line 253: " FeCPs are poorly crystalline in structure and thus comparatively readily soluble". I am not agree with the authors for this generalization. Did you examine the type of corrosion products then you can assume that they are readily soluble yet quantitatively extractable by AA? Even the authors in the same paragraph cited other works that acknowledging difficulties in reducing iron oxides, even under acidic conditions and in the presence of chelating agents!
2. Because most applications of Fe0 are in aerated environment, discussions should include the possible oxidation of ferrous to ferric ions and subsequent mechanisms regarding their back-reduction before determining the intrinsic reactivity of Fe0. This hint must be accounted while comparing between different estimation methods.
Author Response
Comments 1: After the modifications, I think the authors do not reply to the previous points adequately.
Responses 1: Thanks for this remark. Unfortunately, it is not specified which aspects were not adequately addressed. We have added the comments tot he original submission, with some remarks.
Comments 2: Although the enclosed work investigated the suitability of AA method to assign the reactivity of Fe0 specimens, the revised manuscript contains some ambiguous statement,
Responses 1: We are herewith addressing the few named and the original comments of the Reviewer.
Comments 3: 1. Line 253: "FeCPs are poorly crystalline in structure (5 refs. added) and thus comparatively readily soluble (5 refs. added)".
Responses 3: For us this is obvious and do not even need to be referenced. However, we have added four references from the research group of Prof. Gillham (Canada), where Dr. Odziemkowski pioneered investigations on the nature of iron corrosion products (for the ZVI remediation community), using various techniques. We have added a review paper by Dr. Chaves (2005). We have also added referenced for the differential solubility of iron oxides, starting by the historical work of Udo Schwertmann.
Chaves L.H.G. (2005): The role of green rust in the environment: A review. Rev. Bras. Eng. Agríc. Ambient.9, 284–288.
Odziemkowski M.S., Simpraga R.P. (2004): Distribution of oxides on iron materials used for remediation of organic groundwater contaminants - Implications for hydrogen evolution reactions. Can. J. Chem./Rev. Can. Chim. 82, 1495–1506.
Odziemkowski M. (2009): Spectroscopic studies and reactions of corrosion products at surfaces and electrodes. Spectrosc. Prop. Inorg. Organomet. Compd. 40, 385–450.
Ritter K., Odziemkowski M.S., Gillham R.W. (2002): An in situ study of the role of surface films on granular iron in the permeable iron wall technology. Jour. Cont. Hydrol. 55, 87–111.
Ritter K., Odziemkowski M.S., Simpgraga R., Gillham R.W., Irish D.E. (2003): An in situ study of the effect of nitrate on the reduction of trichloroethylene by granular iron. J. Contam. Hydrol. 65, 121–136.
Comments 3: I am not agree with the authors for this generalization. Did you examine the type of corrosion products then you can assume that they are readily soluble yet quantitatively extractable by AA? Even the authors in the same paragraph cited other works that acknowledging difficulties in reducing iron oxides, even under acidic conditions and in the presence of chelating agents!
Responses 4: It is not our generalization, but rather an old knowledge resulting from the evidence that "rust never rests". Despite differences in storage conditions, on the long run, iron corrosion products (FeCPs) is a mixture of iron oxides and hydroxides with different degree of crytallinity. The rest is established knowledge that constitutes the roots of sequential extraction. The idea here is that the fraction (or amount) of FeCPs that can be dissolved in 2 mM AA is dissolved within some few hours (< 1 day). If you change the concentration of AA or change the acidity of the solution, things will change. So this is intuitive and we have referenced Tressier et al., together with other articles more relevant for iron oxides, including Ford (2002).
Comments 5: 2. Because most applications of Fe0 are in aerated environment, discussions should include the possible oxidation of ferrous to ferric ions and subsequent mechanisms regarding their back-reduction before determining the intrinsic reactivity of Fe0. This hint must be accounted while comparing between different estimation methods.
Responses 5: Yes but the originality of the method is that, once FeII is built, it is stabilized by AA and is not further oxidized to FeIII. This is clearly stated in the submission and was already the rationale for the development of the Phen method. The AA method is a further progress because it is cheeped, AA is non-toxic (while Phen is toxic), and atmospheric FeCPs are (at least partly) reduced by AA, Because EDTA forms stable complexes with FeIII, the regression parameters were not optimally determined, and in particular negative b-values were difficult to justify. We feel we have explained this in details already in the original submission. Therefore the concern of this Reviewer is not clear to us.
Comments on the original version
- The effectiveness of Phen method for determining the intrinsic reactivity of Fe0 is relied on the fact that it addressed the forward dissolution of Fe0 with neglecting the contribution from dissolving corrosion-based species. (Phen exhibits no reductive properties for FeIII, thus does address only the FeIII fraction in FeCPs) In the present work, AA induces both the oxidative dissolution of Fe0and the reductive dissolution of atmospheric corrosion products (correct!). Therefore, I think AA test will show an overestimation for the reactivity of Fe0 specimens (Why? First, FeCPs represent less than 1 % of the weighted Fe0. Second, FeIII from FeCPs is reduced within few hours) (did not handle the limitation of EDTA method – while with EDTA, everything is oxidized to FeIII, that is how the limitations are handled, It was clear in the original submission).
2. I wonder why the batch experiments to assign the time-dependent dissolution of Fe0 in AA, Phen and EDTA were not continued for enough time as in previous work (Water 2019, 11, 246). Because the linearity was achieved for experimental points for t > 24 h. All three methods are rooted on the linearity of [Fe] = f(t). This is the reason why, we have given a final protocol for 120 h, although our experiments were performed for 144 h. In preliminary works, using [AA] > 2 mM different observations were made.

Reviewer 2 Report
The comments have been answered. It can be accepted.